# Choroidal Melanocytic Hamartoma

**DOI:** 10.3390/jcm11205983

**Published:** 2022-10-11

**Authors:** Ramesh Venkatesh, Sameeksha Agrawal, Nikitha Gurram Reddy, Rubble Mangla, Naresh Kumar Yadav, Jay Chhablani

**Affiliations:** 1Narayana Nethralaya, Department of Retina and Vitreous, #121/C, Chord Road, 1st R Block Rajaji Nagar, Bangalore 560010, India; 2Medical Retina and Vitreoretinal Surgery, University of Pittsburgh School of Medicine, 203 Lothrop Street, Suite 800, Pittsburg, PA 15213, USA

**Keywords:** choroid, melanin, hamartoma, multimodal imaging

## Abstract

We report on a case series that revealed flat, choroidal lesions on optical coherence tomography (OCT) and on enface MultiColor^®^ (MCI) imaging of the fundus but were not noticeable on clinical examination or conventional color fundus images. This observational study included 12 eyes from 11 patients who had distinct, orange-colored lesions on MCI. Retinal imaging was conducted using conventional color fundus photography and OCT. On the color fundus images and the blue and green reflectance channels of MCI, each of the lesions was difficult to distinguish. On the infrared channel, the lesion was identified as bright white in color and bright orange on the multicolor image. The lesion was identified on OCT as a flat, homogeneous hyperreflective lesion involving the choroid, with an intact overlying retinal pigment epithelium and retinal layers. A comparison of the clinical and imaging features with other known entities led to the conclusion that the lesion was a distinct clinical entity. The presence of melanin in the lesion was confirmed based on the retinal imaging findings and the light absorption properties of melanin. As a result, the lesion was named as ‘choroidal melanocytic hamartoma’. A longer follow-up is required to confirm the benign nature of this clinical entity.

## 1. Introduction

Pigmented or hypopigmented fundus lesions can occur as a result of an increase or decrease in melanin pigment within the melanocytes or melanocytes themselves, involving the retinal pigment epithelium and/or choroid [1]. This can be congenital or acquired as a result of an infection, inflammation, or trauma. They can be flat or elevated morphologically. The known causes of fundus pigmentary disorders include choroidal nevi, congenital hypertrophy of the retinal pigment epithelium, combined hamartoma of the retina and retinal pigment epithelium, isolated choroidal melanocytosis, optic nerve head melanocytoma, uveal melanoma, torpedo maculopathy, amelanotic choroidal nevus, amelanotic choroidal melanoma, acquired hyperplasia of the retinal pigment epithelium, inflammatory conditions such as acute zonal occult outer retinopathy, post-traumatic pigmentary atrophy, and peripapillary atrophy [2]. Fundus photography is used to document, assess, and/or quantify changes in the posterior pole retinal and choroidal lesions. The conventional true color fundus photography (CFP) and pseudo color MultiColor^®^ imaging available on the Spectralis, Heidelberg platform are currently the most commonly used retinal imaging modalities for these purposes [3,4].

We observed a few cases in our clinical practice where MultiColor^®^ imaging identified a bright white lesion on the infrared channel or a bright orange lesion on the multicolor image that was barely visible in clinical examination or documented on conventional CFP. These lesions did not meet the previously discussed criteria for hypopigmented lesions.

We intend to present and analyze the multimodal imaging characteristics of these lesions using the different spectrum and properties of light of what we will call the ‘choroidal melanocytic hamartoma’ in this study.

## 2. Methods

For this study, we included the eyes of patients who had showed a distinct bright orange-colored lesion at the posterior pole on MultiColor^®^ imaging (Spectralis, Heidelberg Engineering, Franklin, MA, USA). The number of such lesions on the multicolor image were noted. The lesion characteristics was documented using other imaging techniques such as conventional digital CFP (Topcon TRC-50Dx, Oakland, NJ, USA) and spectral domain optical coherence tomography (OCT) (Spectralis, Heidelberg Engineering, Franklin, MA, USA). OCT scans were obtained through the lesion and macula using the Spectralis device in all eyes. Macular volumetric assessments were conducted using 512 A-scans per line with a 30° scanning area, 25-line horizontal raster volume scans, and 12-line radial scans centered on the fovea. Through the lesion, dense scans using the enhanced depth imaging mode were performed. All of the OCT scans were analyzed by a single observer (SAM) to check for the exact depth of the lesion and for changes in the structures above and below the lesion. Patient details including age, gender, best-corrected visual acuity and anterior segment findings, and intraocular pressure were noted. Patients with poor image quality, which prevented proper description of the lesions, and eyes with far peripheral lesions that were only partially visible in any imaging modality were excluded. None of the patients had fluorescein or indocyanine green angiography, or ocular ultrasound to further image these lesions. The hospital’s institutional review board and ethics committee granted all of the necessary permissions for a review of the patients’ charts and images as well as publication.

## 3. Results

The study included 12 eyes from 11 patients who had distinct orangish lesions on the multicolor images. The study included eight males and three females. The patients’ average age was 48.92 years (range: 22–75 years). Ten patients had unilateral involvement, while one patient had bilateral involvement. Systemic association was noted in two patients: one with neurofibromatosis type 2 and one with tuberous sclerosis. Multiple lesions were found in the eyes of these two patients. Anterior segment examination was normal in all eyes except in one patient of neurofibromatosis type 2 who showed Lisch nodules on the iris in both eyes. In total, 29 lesions were identified in the study (one per eye in nine cases, four in one eye in one case, and eight each in both eyes in one case). The visual acuity in these eyes ranged from 6/6 to 6/18.

Similarities were noted in the description of the lesions on MultiColor^®^ imaging, OCT, and CFP in all of the eyes. The lesions were poorly distinguished on the conventional CFP and on the blue and green reflectance channels of MultiColor^®^ imaging. The lesions were well-defined on the infrared channel and were seen as bright-white in color and hence seen on the enface composite multicolor image as a bright-orange-colored lesion. The lesion on OCT was a flat, homogeneous hyperreflective lesion involving the choroid, primarily the inner choroid, with the overlying retinal pigment epithelium and retinal layers intact. Over the lesion, there was no elevation of the retinal pigment epithelium, subretinal fluid, orange pigment, or drusen. Due to shadowing, the choroidal vessels beneath the lesion were not visible when compared to the choroidal vessels surrounding the lesion (Figure 1 and Figure 2). The mean follow-up duration following the identification of the lesion ranged from 1 to 8 years. No change in the lesion size, depth, or extent was noted during the follow-up period in any case.

## 4. Discussion

In this study, we identified fundal lesions at the posterior pole, which were overtly noticeable only with the higher wavelength infrared reflectance images, and as a result, on the multicolor image. The lesions were not visible with the shorter wavelength blue and green reflectance images. To the best of our knowledge, such lesions have not been previously described in the literature. In this paper, we intend to compare these findings with other differential diagnoses and emphasis on the role of MultiColor^®^ imaging in identifying such lesions.

MultiColor^®^ imaging captures three retinal reflectance images at the same time using three different lasers (blue—488 nm, green—515 nm, and infrared—815 nm), allowing for the analysis of changes at various retinal and choroidal levels. These images were combined to create a composite enface multicolor image [5]. A head-to-head comparison between the conventional color fundus and multicolor images for identifying different retinal and choroidal diseases showed the superiority of MultiColor^®^ imaging for surface lesions such as the epiretinal membrane and macular pucker, vascular lesions such as retinal vascular occlusions, and diabetic macular edema and optic nerve head lesions such as disc edema and optic nerve head melanocytoma over the conventional color fundus photograph [6,7,8,9,10,11].

In this series, the lesion was mainly evident with the infrared wavelength light. This, along with the OCT features, suggest that the lesions are located deep in the choroid underneath the retinal pigment epithelium. Between the retina and sclera, the choroid is made up of a pigmented stroma (choroidal melanocytes) and a dense network of blood vessels (choriocapillaris, Sattler’s and Haller’s layers) [12]. The infrared spectrum of light is poorly absorbed by melanin in the choroidal stroma [13,14]. Increased melanin within melanocytes and/or increased melanocytes reflect light from the near-infrared spectrum (780 nm—1000 nm) and show increased reflectivity with poor visibility of the underlying structure in areas of increased pigmentation [15]. From the description of the MultiColor^®^ imaging features in the current series of cases, it is likely that the lesion consists of an accumulation of choroidal melanocytes. As a result, pathologies characterized by increased pigmentation on the choroid such as choroidal nevus, choroidal melanoma, and isolated choroidal melanocytosis must be considered [16,17,18,19].

Choroidal nevus is a brown or tan mass with a round or oval shape that is located deep beneath the retinal pigment epithelium. Choroidal nevi are distinguished by variable thickness, distinctive OCT patterns, and distortions on fluorescein angiography [16,20,21]. In our series, this flat lesion was scarcely seen on clinical examination or documented on a conventional color fundus photograph as a pigmented lesion. Additionally, there was no associated retinal pigment epithelium elevation or presence of subretinal fluid or drusen. The OCT patterns were not typical of choroidal nevi, as previously described by Jonna et al. [20].

Uveal melanomas commonly involve the choroid, and choroidal melanomas are usually described as broad based and elevated pigmented tumors underneath the retina [18,22]. In this series, the lesions were not pigmented on examination and were seen as flat lesions on the OCT scans. Moreover, choroidal melanomas are commonly seen in the elderly (range: 50–80 years) while in our series, the lesions were also seen in the younger age group patients. Thus, choroidal nevus and choroidal melanoma are an unlikely diagnosis in our series of cases.

In 2006, Augsburger described a new clinical entity that appeared as flat, diffuse, and dark brown in color on the fundus and termed it as isolated choroidal melanocytosis [23]. Although the current case series showed that the OCT and multicolor imaging features were similar to what we described in our previous publication on isolated choroidal melanocytosis, the clinical description did not match that described by Augsburger et al. [17,23,24]. Thus, isolated choroidal melanocytosis also seems to be an unlikely diagnosis.

In our series, we noted a normal looking fundus with a collection of melanocytic cells in the choroid on the OCT and MultiColor^®^ imaging, not showing any change during follow-up [25]. This could be compared to the Lisch nodules seen in neurofibromatosis type 2 and other phacomatoses [26]. In addition, two patients in our series had associated neurofibromatosis type 2 and tuberous sclerosis. Hence, we described this entity as choroidal melanocytic hamartoma. The differentiating features between these three entities and choroidal melanocytic hamartoma are described in Table 1 and Figure 3.

To summarize, a case series of flat, orange-colored lesions noted in the posterior pole on MultiColor^®^ imaging, poorly distinguished on clinical examination, and located deep in the choroid, as seen on OCT, has been described for the first time. Multimodal imaging supports the classification of these lesions as melanocytic hamartomas of the choroid. This entity can be seen alone or in conjunction with phacomatoses such as neurofibromatosis type 2 and tuberous sclerosis. A longer follow-up of these lesions is required to confirm the benign nature of this clinical entity.

## Figures and Tables

**Figure 1 jcm-11-05983-f001:**
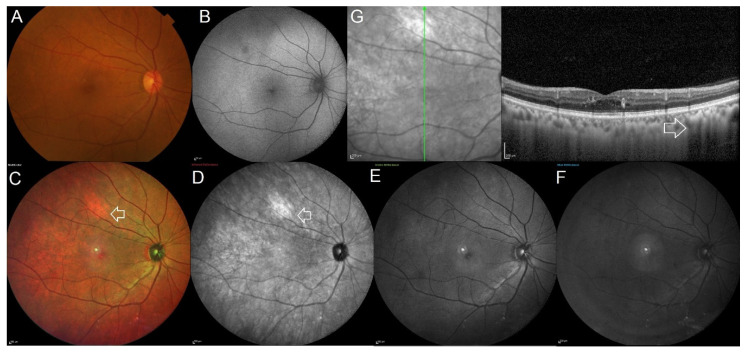
Multimodal imaging in a patient showing isolated choroidal melanocytic hamartoma in the right eye. (**A**) Conventional color fundus image of the posterior pole obtained using the Topcon TRC 50 Dx fundus camera. The color fundus image looks normal and does not identify any significant pathology. (**B**) The 55° blue wavelength autofluorescence image appears normal. (**C**–**F**) A bright orange colored lesion is noted along the superotemporal vessel on the 55° multicolor image (white arrow) and is seen as a bright white hyperreflective lesion on the infrared reflectance channel (white arrow). The lesion is not visible on the short wavelength green and blue reflectance channels. (**G**) Enhanced depth imaging optical coherence tomography scan passing through the fovea and lesion showed a normal foveal contour and increased reflectivity in the choroid with poor visibility of the underlying structure at the area of the lesion (white arrow).

**Figure 2 jcm-11-05983-f002:**
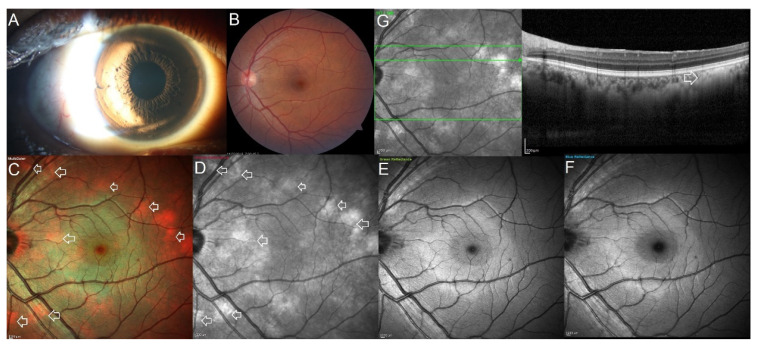
Multimodal imaging in a patient with neurofibromatosis type 2 showing multiple isolated choroidal melanocytic hamartomas in the left eye. (**A**) Slit-lamp photograph with oblique illumination shows the Lisch nodules on the iris surface. (**B**) The 50° conventional color fundus photograph of the left eye looks normal and does not identify any pathological lesions. (**C**–**F**) The 30° multicolor image shows multiple choroidal melanocytic hamartomas spread across the posterior pole as bright orange-colored lesions (white arrows). On the high wavelength infrared reflectance channel, the lesions appear as white hyperreflective lesions (white arrows). The lesions are not visible on the short wavelength green and blue reflectance channels. (**G**) Enhanced depth imaging optical coherence tomography scan passing through one of the lesions shows increased reflectivity in the choroid with poor visibility of underlying structure at the area of the lesion (white arrow).

**Figure 3 jcm-11-05983-f003:**
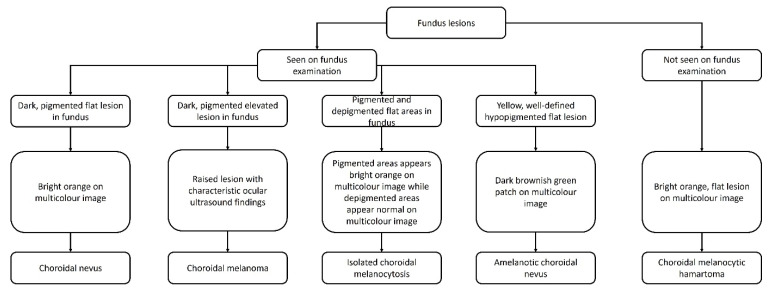
Flow chart depicting the clinical and imaging findings of different differential diagnosis.

**Table 1 jcm-11-05983-t001:** Comparison between the differential diagnoses.

	Choroidal Nevus	Choroidal Melanoma	Isolated Choroidal Melanocytosis	Choroidal Melanocytic Hamartoma (Our Case Series)
Age	Middle-age to elderly	Peaks at middle-age to elderly	Any age from infants to elderly	Young adults to elderly
Sex distribution	M = F	M > F	M < F	M > F
Symptoms	Asymptomatic	Usually asymptomatic but can present with vision loss, scotoma, photopsia or floaters.	Asymptomatic	Asymptomatic
Clinical features	Usually round, flat, small and grey colored lesion. 10% amelanotic.	Variable shape and size, often dark grey and irregular and raised lesion. 15% amelanotic.	Large, flat pigmented lesion with either focal or annular distribution with adjacent hypopigmented or depigmented lesion.	Poorly distinguished on clinical examination.
MultiColor^®^ imaging	Salmon patch appearance. Sometimes can show variable presentation.	Lesion features like border, drusen and halo are similar to that seen on clinical examination. Size of the lesion appears lesser compared to clinical conventional fundus photograph.	Area of choroidal melanocytosis appears bright orange in color.	More obvious and distinct as a bright-orange colored lesion.
Optical coherence tomography features	Flat or dome shaped with diffuse hyperreflectivity, posterior shadowing, compressed choriocapillaris and rarely associated with drusen, subretinal fluid and/or choroidal neovascularization.	Smooth, dome-shaped destruction of the retinal pigment epithelium and outer retina, as well as associated subretinal fluid and “shaggy” photoreceptors.	There is increased hyperreflectivity in the choroid with poor visibility of underlying choroidal structures in the presence of choroidal melanocytosis. There is an obvious increase in choroidal thickness in the surrounding non-pigmented area, with poor visibility of choroidal architecture. There is no associated RPE elevation, drusen, orange pigment, or subretinal fluid.	Hyperreflectivity in the choroid with poor visibility of the choroidal structures beneath. The choroid around the lesion appears to be normal. There is no associated RPE elevation, drusen, orange pigment, or subretinal fluid.
Growth	Slow and minimal	Rapid	Slow and minimal	No growth

Abbreviations: M—male; F—female; RPE—retinal pigment epithelium.

## Data Availability

The datasets used and/or analyzed during the current study are available from the corresponding author on reasonable request.

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
