# Peer review of "Choroidal Melanocytic Hamartoma"

_jcm, 2022, doi:10.3390/jcm11205983_

Round 1

Reviewer 1 Report

This is an article entitled “Choroidal melanocytic hamartoma (jcm-1937162)” which demosntrates a case series showing choroidal lesions on optical coherence tomography and on Multicolour imaging but hardly seen on clinical examination and conventional colour fundus photography.

The abstract and the title are not going along. Either the title or the abstract should be changed.

Abstract

-          - Confusing.

Introduction

-          - Should be enlarged and rewritten.

Methods

-          - Please name all the tools you used in a detailed manner, type, the place where it was produced etc.

Results

-          - Did you perform florescein angiography etc as well? It would be better to also see the lesions on FFA or maybe ICG-A.

-        -   Did you perform ultrasound?

Discussion

-       -  Ok.

References

-         -  Ok.

Figures

-       -   Ok.

Reviewer 2 Report

This manuscript is an excellent description of choroidal lesions in 12 eyes of 11 patients not being detectable by ordinary fundus examination or conventional fundus photography. They were identified on the infrared channel of Multicolor imaging as white lesions or as orange lesions on the multicolor image. On OCT the lesions were hyperreflective and situated in the choroid. The authors describe these lesions as choroidal melanocytic hamartomas after comparing them to well known pathologies of the choroid (nevus, melanoma, and melanocytosis – Table 1). Both Figures are very instructive and of excellent quality. In summary an interesting cases series with high clinical impact (advice to use multicolor imaging).

Just one comment: Line 148: 780 nm – 1 mm. shouldn´t it be 1 µm?

Reviewer 3 Report

The manuscript entitled "Choroidal melanocytic hamartoma" is based on an interesting case series of 12 eyes in 11 patients with choroidal melanocytic hamartoma. This study provides interesting insights into this type of retinal lesion that is typically difficult to detect in fundus examination, or simply overlooked and not given the importance needed. 

The topic adds to the current literature and is of clinical interest, especially considering the rarity and difficulty in diagnosing this type of lesion. The authors should be commended for the interesting examples, and appropriate OCT scans and fundus photos, which can be helpful to  clinicians in the detection and differential diagnosis of choroidal melanocytic hamartoma. The paper is well planned, and the explanations, OCT scans, photos, and table reporting the findings can be of clinical use.

I have only one comment that could render the paper more practical in clinical. The authors should provide more information and appropriate references on differential diagnosis, chances of malignancies in these lesions, and maybe a simple flowchart on what should be done and follow-up suggestions that could be useful in a routine clinical setting when faced with patients with these retinal lesions.

Reviewer 4 Report

The abstract should be clear and sound… e.g.:

Page 1 line 34:

To report a case series showing choroidal lesions on optical coherence tomography (OCT) and on Multicol- 34 our® (MCI) imaging …: please specify of what?

Page 2 line 52: please insert post traumatic retinal atrophy.

Round 2

Reviewer 1 Report

Suggested revisions are made.